# Management of Uveal Melanoma: Updated Cancer Care Alberta Clinical Practice Guideline

**Ezekiel Weis** [1,2,*], **Brae Surgeoner** [3], **Thomas G. Salopek** [4], **Tina Cheng** [5], **Martin Hyrcza** [6], **Xanthoula Kostaras** [3], **Matthew Larocque** [7], **Greg McKinnon** [8], **John McWhae** [9], **Geetha Menon** [7] , **Jose Monzon** [5], **Albert D. Murtha** [10], **John Walker** [11] and **Claire Temple-Oberle** [8]

1   Department of Ophthalmology, University of Alberta, Edmonton, AB T6G 2E1, Canada
2   Department of Surgery, University of Calgary, Calgary, AB T2N 1N4, Canada
3   Cancer Care Alberta, Calgary, AB T2S 3C3, Canada; brae.surgeoner@ahs.ca (B.S.);
    xanthoula.kostaras@ahs.ca (X.K.)
4   Division of Dermatology, Department of Medicine, University of Alberta, Edmonton, AB T6G 2G3, Canada;
    tsalopek@ualberta.ca
5   Tom Baker Cancer Center, Division of Medical Oncology, Department of Oncology,
    Calgary, AB T2N 4N2, Canada; tina.cheng@ahs.ca (T.C.); jose.monzon@ahs.ca (J.M.)
6   Laboratory Medicine, Department of Pathology, Arnie Charbonneau Cancer Institute, University of Calgary,
    Calgary, AB T2N 1N4, Canada; martin.hyrcza@albertaprecisionlabs.ca
7   Division of Medical Physics, Department of Oncology, Faculty of Medicine and Dentistry, University of
    Alberta, Edmonton, AB T6G 1Z2, Canada; matthew.larocque@ahs.ca (M.L.); geetha.menon@ahs.ca (G.M.)
8   Division of Surgical Oncology, University of Calgary, Calgary, AB T2N 1N4, Canada;
    greg.mckinnon@ahs.ca (G.M.); claire.temple-oberle@ahs.ca (C.T.-O.)
9   Departments of Surgery and Oncology, University of Calgary, Calgary, AB T2N 1N4, Canada;
    john.mcwhae@ahs.ca
10  Division of Radiation Oncology, Cross Cancer Institute, University of Alberta,
    Edmonton, AB T6G 1Z2, Canada
11  Division of Medical Oncology, Cross Cancer Institute, University of Alberta, Edmonton, AB T6G 1Z2, Canada;
    john.walker2@ahs.ca
*   Correspondence: ezekiel.weis@ahs.ca

**Abstract:** Objective: The purpose of this guideline update is to reassess and update recommendations in the prior guideline from 2016 on the appropriate management of patients with uveal melanoma. Methods: In 2021, a multidisciplinary working group from the Provincial Cutaneous Tumour Team, Cancer Care Alberta, Alberta Health Services was convened to update the guideline. A comprehensive review of new research evidence in PubMed as well as new clinical practice guidelines from prominent oncology groups informed the update. An enhancement in methodology included adding levels of evidence and strength of recommendations. The updated guideline was circulated to all members of the Provincial Cutaneous Tumour Team for review and endorsement. Results: New and modified recommendations address provider training requirements, diagnostic imaging for the detection of metastases, neo-adjuvant pre-enucleation radiotherapy, intravitreal anti-vascular endothelial growth factor agents for radiation retinopathy, genetic prognostic testing, surveillance following definitive local therapy, and systemic therapy for patients with metastatic uveal melanoma. Discussion: The recommendations represent evidence-based standards of care agreed to by a large multidisciplinary group of healthcare professionals.

**Keywords:** uveal melanoma; ocular melanoma; choroidal melanoma; iris melanoma; melanoma; ophthalmology; practice guidelines

## 1. Introduction

Melanoma of the uveal tract (i.e., iris, ciliary body, and choroid), also referred to as 'ocular melanoma', accounts for 5% of all melanomas and occurs at an incidence rate of about six cases per million person-years [1,2]. Use of 'ocular melanoma' is discouraged because it

does not differentiate uveal melanoma from melanoma arising from the conjunctiva and eyelid skin. Risk factors for uveal melanoma include light eye and skin colour, cutaneous and iris nevi and freckles, an inability to tan, BAP1 tumour predisposition syndrome, and exposure to arc welding and suntan beds [3–8]. Uveal melanoma is the most common primary intraocular malignancy, and the uveal tract is the second most common location for melanoma after the skin [2]. Uveal melanoma arises from melanocytes but is distinct from cutaneous melanoma in that it has different molecular drivers, metastatic patterns, and tumour-immune microenvironment [9–11].

Uveal melanoma carries a 50% long-term survival rate [12]. Adverse prognostic factors for survival include altered gene expression profiles, specific mutations, patient age, maximal basal tumour diameter, location, cytogenetic alterations, ciliary body involvement, extrascleral tumour extension, and epithelioid cell type [13–15]. Despite advancements in local treatments with improvements in eye-sparing techniques, improvement in survival has not yet been documented [8].

The timely management of uveal melanocytic lesions including small flat lesions is vitally important. Delays in referral and treatment may result in both complete loss of the eye (i.e., enucleation) and/or life (i.e., metastasis). Research shows that earlier treatment, allowing for treatment of a smaller lesion, portends improved survival [16]. Furthermore, waiting for observation of growth in small lesions identified as high risk by an ophthalmologist can increase the risk of metastasis by eight times [17]. Therefore, even melanocytic lesions ≤3 mm thick without any documentation of growth can be offered treatment [17,18].

The purpose of this guideline is to reassess and update recommendations made in the previously published guideline from 2016 on the appropriate management of patients diagnosed with uveal melanoma [19]. Other 'ocular' melanomas arising in the conjunctiva, the eyelid, and the orbit are excluded from these guidelines.

## 2. Materials and Methods

### 2.1. Research Questions

(i)     For patients with uveal melanoma, which staging investigations are required at baseline?
(ii)    How should patients with uveal melanoma, including patients with metastatic or recurrent disease, be managed?
(iii)   What is the recommended surveillance strategy for patients diagnosed with uveal melanoma?

### 2.2. Literature Search

PubMed was searched from 1 January 2014 (end date of earlier literature review) until 2 March 2021, using the Medical Subject Heading (MeSH) term 'Uveal Neoplasms'. The search strategy included clinical trials, meta-analyses, observational studies, practice guidelines, and systematic reviews that included individuals 19 years old and older, and that were published in English. Reference lists of key publications were also searched for relevant citations. Clinical practice guidelines on uveal melanoma were located using the PubMed National Library of Medicine (Bethesda, Rockville, MD, USA) and ECRI Guidelines Trust® (Plymouth Meeting, PA, USA) databases, as well as known guideline developer websites. As per Cancer Care Alberta's Guideline Methodology Handbook, only guidelines published in the past five years were considered for inclusion [20]. The search strategy, review of results, and synthesis of relevant studies was completed by a methodological expert from Cancer Care Alberta's Guideline Resource Unit (B.S.).

### 2.3. Internal Review

Evidence tables were reviewed by members of the original Working Group, consisting of two ophthalmologists, two surgical oncologists, one radiation oncologist, three medical oncologists, one dermatologist, one pathologist, and two radiologists. Over the course of several Working Group meetings led by the lead author (E.W.), the original guideline was

reviewed, recommendations revised as needed considering the new evidence, and level of evidence and strength of recommendations added. The definitions for the level of evidence and strength of recommendations listed in Table 1 are adapted from the Infectious Diseases Society of America and the European Society of Medical Oncology [21,22]. Finally, the revised draft guideline was circulated to all members of the Provincial Cutaneous Tumour Team for review and endorsement.

**Table 1.** Description of level of evidence and strength of recommendations for this guideline.

| Level | Description of Evidence |
|---|---|
| I | At least one large randomized controlled trial (RCT) of good methodological quality with low potential for bias or meta-analyses of RCTs without heterogeneity. |
| II | Small phase III RCT or phase II RCT or large phase III RCT with potential bias or meta-analyses including RCTs with heterogeneity. |
| III | Prospective cohort studies or post/ad hoc analyses of RCTs. |
| IV | Retrospective cohort studies or case-control studies. |
| V | Studies without a control group or expert opinion. |
| Grade | Description of Strength of Recommendation |
| A | Strongly recommended; strong evidence for efficacy with a substantial clinical benefit. |
| B | Generally recommended; strong or moderate evidence for efficacy but with a limited clinical benefit. |
| C | Optional; insufficient evidence for efficacy or benefit does not outweigh the risks/disadvantages. |
| D | Generally not recommended; moderate evidence against efficacy or for adverse outcomes. |
| E | Never recommended; strong evidence against efficacy or for adverse outcomes. |

*2.4. Ad Hoc Recommendation Update*

In 2023, during the preparation of this manuscript, the Working Group decided to update the recommendation for the treatment of metastatic uveal melanoma in response to new evidence concerning a drug called tebentafusp (a novel bispecific protein comprised of a soluble T cell receptor fused to an anti-CD3 immune-effector function) that was previously regarded as experimental during the initial literature review and internal review process. Additionally, the Working Group considered new data about the effectiveness of percutaneous hepatic perfusion (PHP) with melphalan for patients with uveal melanoma liver metastases. The Working Group agreed that while PHP is an attractive option, its application is limited by the availability of the procedure to specialized cancer centres and immature overall survival data.

**3. Results**

*Literature Search Results*

The literature search identified 101 publications in PubMed that required abstract screening for inclusion in the updated guideline. Additionally, seven clinical practice guidelines or peer-reviewed cancer information summaries were identified. From these results, a total of 33 new publications informed this guideline update, including eight randomized controlled trials. Evidence tables are available upon request.

**4. Clinical Practice Guideline**

*4.1. Recommendations for Diagnosis and Work-Up*

- An opthalmologist trained in all aspects of care (i.e., medical, oncologic, surgical, radiation and laser therapy) should evaluate all intraocular malignancies and indeterminate lesions to determine appropriate follow-up and/or treatment. (Level of evidence: V [23], Strength of recommendation: B).
- Complete history, including opthalmic and medical history. (Level of evidence: V [24], Strength of recommendation: B).

- Complete opthalmic examination and funduscopy, including a baseline fundus photograph of sufficient quality, and an objective height assessment of all melanocytic lesions. (Level of evidence: V [24], Strength of recommendation: B).
- Ocular ultrasonography (A/B-scan, ultrasound [US], ultrasound biomicroscopy [UBM]) by a certified opthalmic ultrasonographer or opthalmologist with US training. (Level of evidence: V [23,24], Strength of recommendation: B).
- Ancillary ocular studies, in cases where opthalmic examination is inconclusive (e.g., media opacity). (Level of evidence: V [23,24], Strength of recommendation: B).
- Staging work-up to rule out metastases for patients diagnosed with uveal melanoma. (Level of evidence: V [24], Strength of recommendation: B).

### 4.1.1. Qualifying Statements

A-scan US demonstrates initial prominent spike followed by low-to-medium internal reflectivity or a decrescendo pattern and can be used to measure tumour height [25]. B-scan US allows for tumour measurement and characteristics including solidity/hollowness, vascularity, shape, and extrascleral extension [25]. UBM provides high-resolution imaging of the anterior segment of the eye for visualization of ciliary body and iris tumours [25–27].

Fluorescein and/or indocyanine green angiography of the retina and choroidal vascularity is helpful as an ancillary study in select cases (requires clear media for visualization). Orbital/ocular computed tomography (CT) and magnetic resonance imaging (MRI) are generally not required in the diagnosis work-up, unless other examinations are inconclusive [28].

The common lesions on the differential diagnosis for uveal melanoma include freckles, nevus, Lisch nodules, neovascular age-related macular degeneration, congenital hypertrophy of the retinal pigment epithelium (CHRPE), retinal pigment epithelium (RPE) adenoma, choroidal hemangioma, hemorrhagic detachment of the choroid or retina (peripheral exudative hemorrhagic chorioretinopathy), melanocytoma, metastasis to the eye from another location, and choroidal osteoma [23].

Staging work-up to rule out metastases for patients diagnosed with uveal melanoma should include blood work (i.e., complete blood count, liver function tests [LFTs], and lactate dehydrogenase [LDH]), as well as appropriate use of diagnostic imaging. Individuals who would be a candidate for treatment of metastatic disease should undergo a whole-body positron emission tomography (PET)/CT scan, if available. Alternatively, a CT scan of the chest, abdomen, and pelvis should be done. Gadolinium enhanced MRI may be indicated in select patients. If metastasis is suspected, the patient should be referred to a tertiary cancer centre. The treating surgeon should decide on the appropriateness of staging investigations, considering the fitness of the patient for subsequent treatment(s) of metastatic disease (should it be found).

### 4.1.2. Key Evidence

Staging is guided by the American Joint Committee on Cancer (AJCC) system for uveal melanoma [29]. Whole-body PET-CT scan has demonstrated good sensitivity (35–100%) and positive predictive value (88–100%) [30–32], while MRI has shown the highest sensitivity (67–92%) [31–33].

Controversy exists around whether baseline imaging should be performed in this population, based on the historical premise that metastases cannot be treated and the yield of positive findings at presentation is low. More than half of the patients have CT abdomen findings that require further investigation, the majority of which are false positives [34]. Only 3.3% have definitive metastasis at staging. An international, registry-based retrospective data analysis of patients presenting with stage IV uveal melanoma found that 6% of the uveal melanoma with metastasis at initial presentation belonged to subcategory T1a, most often detected by whole-body PET/CT [35].

The diagnosis and work-up recommendations align with other current clinical practice guidelines published by the National Comprehensive Cancer Network (NCCN) [24] and the National Cancer Institute (NCI) [23].

### 4.1.3. Justification

The diagnosis of uveal melanoma is difficult for the non-specialist [28,36]. Experienced ophthalmologists with a practice focus in oncology can diagnose uveal melanoma based primarily on funduscopy and US (i.e., without biopsy) with 98% accuracy [37]. Treatment options for uveal melanocytic lesions involve medical, laser, extraocular and intraocular surgery, radiotherapy (RT), and other eye-sparing treatment modalities [38]. Observation versus treatment discussions, especially for small melanocytic lesions, often require balancing treatment-associated complications with the risk of observation and its potential to alter survival. Furthermore, to reduce the risks of local relapse and to reduce the extent of visual loss following eye-sparing treatments, adjuvant medical, laser, and complicated surgical treatments often need to be administered [39–42]. For these reasons, the Working Group recommends that the provider be fully trained in all treatment areas (i.e., medical, surgical, RT, laser treatments of the eye, and cancer care) to safely follow, discuss, and treat all indeterminate (uveal melanocytic lesions that have not demonstrated growth) and malignant intraocular lesions.

### 4.2. Recommendations for Primary Management

- Small choroidal melanomas/indeterminate melanocytic lesions (<3 mm thick) (i.e., nevi, indeterminate melanocytic lesions, and melanomas) should be evaluated based on risk factors for growth and the associated risk of visual loss with treatment. Most tumours without risk factors should be observed until growth is documented. Once growth is documented, the lesion is labeled a melanoma and should be treated (Level of evidence: III [43–45] IV [17,46], Strength of recommendation: B).
- Medium (3–12 mm thick and <16 mm in maximal basal dimension) melanocytic choroid tumours should be offered definitive treatment such as ocular brachytherapy (Level of evidence: I [36,47,48], Strength of recommendation: A).
- Large (>12 mm thick) melanocytic choroid tumours are offered enucleation (Level of evidence: I [49], Strength of recommendation: B) or brachytherapy (Level of evidence: III [50] IV [51,52], Strength of recommendation: C) if adequate dosing can be achieved. Neo-adjuvant pre-enucleation radiation is not recommended (Level of evidence: I [53], Strength of recommendation: E).
- Ciliary body lesions <12 mm thick without extensive circumferential growth pattern should be treated with brachytherapy (Level of evidence: brachytherapy IV [54,55], Strength of recommendation: C)
- Iris lesions should be observed for growth before offering treatment. (Level of evidence: IV [56], Strength of recommendation: C).

### 4.2.1. Qualifying Statements

Observation is typically reserved for indeterminate lesions but may be acceptable for select patients with small melanoma (i.e., <3.0 mm apical height and <10.0 mm basal diameter) under the direction of an ocular oncologist [23]. Most patients selected for observation present with a low-grade tumour and have multiple comorbidities, monocular status, and/or advanced age and already carry a limited expected survival [23].

Risk factors for future growth of indeterminate lesions include tumour thickness > 2 mm, subretinal fluid, symptoms of visual acuity loss to 20/50 or worse, orange pigment, hollow acoustic density, and tumour largest basal diameter >5 mm [57]. If 1–2 of these risk factors are present, close observation, biopsy, or treatment should be considered. High-risk lesions (≥3 risk factors) are often offered treatment, biopsy, or close observation based on discussion with the patient about visual loss, because the risk of future growth is greater than 50% [58]. When indicated, treatment is most commonly ocular brachytherapy [59,60].

Even after controlling for gene expression profiling (GEP), tumour size has been found to be an independent predictor of metastasis at 5 years [61,62]. While brachytherapy is recommended for medium-sized melanocytic choroid tumours, patients who cannot attend follow-up visits may choose enucleation. Many centres offer enucleation for large melanocytic choroid tumours because of the risk of severe vision loss and neovascular glaucoma secondary to radiation complications. However, brachytherapy may be offered to patients with contralateral vision loss or who refuse enucleation.

Both ciliary body and iris lesions are amenable to surgical excision in select cases (iridocyclectomy and iridectomy, respectively) [23,63,64]. Iris and ciliary body lesions are also amenable to brachytherapy if treatment is required [23,65].

### 4.2.2. Key Evidence

The seminal Collaborative Ocular Melanoma Study (COMS) small tumour observational study reported on mortality [45] and factors predictive of growth [44]. Of tumours labeled "small choroidal melanomas" initially managed by observation, 21% demonstrated growth by 2 years and 31% by 5 years [44]. In otherwise healthy patients with an average age of 60 years, and without a previous diagnosis of malignant disease, there was a low risk of dying within 5 years [45]. Several retrospective studies [17,46] and one prospective study [43] have been published that suggest patients with small indeterminate lesions who are carefully selected by an ophthalmologist may be observed for tumour growth before initiating treatment without adversely affecting survival.

At five years, local treatment failure and radiation complications requiring enucleation were relatively infrequent events (10.3% and 5%, respectively) after iodine-125 ($^{125}$I) plaque brachytherapy in the COMS randomized trial for choroidal melanoma measuring 2.5 to 10.0 mm in apical height and no more than 16.0 mm in longest basal dimension [47]. Comparisons of $^{125}$I brachytherapy with enucleation to treat choroidal melanoma failed to identify clinically or statistically meaningful difference in mortality rates between treatment arms for up to 12 years after treatment [36,48]. Five-, 10-, and 12-year rates of death with histopathologically confirmed melanoma metastasis were 10%, 18%, and 21%, respectively, in the $^{125}$I brachytherapy arm and 11%, 17%, and 17%, respectively, in the enucleation arm.

Traditionally, large tumours have been treated with enucleation. A COMS trial of pre-enucleation radiation of large choroidal melanoma reported a 5-year survival rate of 57% in patients treated with enucleation alone, higher than the 50% originally projected [49]. However, ten-year rates of death with histopathologically confirmed melanoma metastasis in the COMS randomized trial of pre-enucleation radiation of large choroidal melanoma were 45% in the pre-enucleation radiation arm and 40% in the enucleation-alone arm [53].

There are no randomized data comparing the efficacy of brachytherapy and enucleation for patients with large tumours. However, a retrospective, nonrandomized comparative trial evaluating the safety and efficacy of $^{125}$I plaque brachytherapy for large uveal melanomas reported that the Kaplan–Meier estimate for melanoma-specific survival was 65% at 5 years [51]. The corresponding estimate for local tumour recurrence was 6% and for major cosmetic abnormality it was 38% [51]. This reported 5-year melanoma-specific mortality was to be comparable with that reported for enucleation in other studies with predominantly large uveal melanomas [13,49].

In a retrospective study of only patients with ciliary body melanoma treated with plaque brachytherapy ($^{125}$I, ruthenium-106 [$^{106}$RU], cobalt-60 [$^{60}$Co], and iridium-192 [$^{192}$Ir]), the 5-year local control rate was 92% [54]. However, the 5-year metastasis rate remained significant at 28% and the 5-year melanoma death rate was 22%. In a retrospective study of patients with iris tumours whose lesions were excised ($n = 36$) or observed ($n = 249$), 89% of the excised tumours were found to be malignant melanomas on histopathologic examination [56]. In the observed group, the actuarial 5-year rate of lesion enlargement was 6.5%. Of the ten lesions that enlarged, six were excised and evaluated histopathologically. Five of the six lesions were malignant melanomas on histopathologic examination. Largest basal tumour diameter was the only clinical variable strongly predictive of lesion enlargement.

*4.3. Recommendations for Adjuvant Local Therapy*

- If margins are positive or indeterminate after resection, adjunctive plaque brachytherapy of the surgical margins may be offered. (Level of evidence: IV [66,67], Strength of Recommendation: C)
- Transpupillary thermotherapy (TTT), diode laser hyperthermia, may be offered to patients with choridal lesions following brachytherapy to reduce the risk of local recurrence (LR). (Level of evidence: III [39] IV [68], Strength of Recommendation: C)
- Intravitreal anti-vascular endothelial growth factor (VEGF) agents can be offered to prevent and/or reduce the severity of radiation retinopathy and its associated visual loss. (Level of evidence: ranibizumab II [69,70] bevacizumab III [71], Strength of Recommendation: C)

### 4.3.1. Qualifying Statements

The Working Group acknowledges that TTT may also be used as primary treatment for medium-risk nevi in select cases. However, used in this setting TTT has been associated with a relatively high rate of LR, especially when the tumour abuts the optic nerve and overhangs the optic disc [72,73]. Therefore, TTT is not recommended as monotherapy for uveal melanoma in the standard case. Additionally, the Working Group acknowledges that TTT is used in some centres to treat marginal recurrence post-brachytherapy [74,75].

### 4.3.2. Key Evidence

A prospective noncomparative interventional case series examined tumour control and treatment complications following plaque brachytherapy combined with TTT for choroidal melanoma [39]. TTT was applied in three sessions. At 5-year follow-up, recurrence was only 3%. Treatment-related complications at 5 years included papillopathy (38%), maculopathy (18%), macular retinal vascular obstruction (18%), and vitreous hemorrhage (18%). A retrospective study looked at supplemental TTT performed in uveal melanoma patients at high risk for LR following $^{125}$I episcleral plaque brachytherapy with US-guided plaque localization and reported similar LR [68]. High-risk features in this study were described as juxtapapillary tumour location or for plaque tilt > 1 mm at plaque removal. After a median follow-up of 45.9 months, LR was detected in only 3.6% of patients. One or two TTT sessions were required in most patients.

The RadiRet study, a phase II clinical trial, compared the efficacy and safety of intravitreal ranibizumab with panretinal photocoagulation (PRP) in radiation retinopathy secondary to radiation of uveal melanoma [70]. Results showed a statistically significant improvement in visual acuity and clear superiority of ranibizumab compared to laser treatment up to 26 weeks, but this effect disappeared at week 52 after completion of intravitreal treatment. A second phase II randomized clinical trial assessing the efficacy of intravitreal ranibizumab injections for radiation retinopathy-related macular edema found that monthly ranibizumab injections led to a 7% improvement in best-corrected visual acuity at 1 year [69]. This gain in visual acuity was statistically significant compared to the monthly injection with targeted PRP group and the as-needed injections with targeted PRP group. Another small prospective study found that bevacizumab treatment was followed by reductions in retinal hemorrhage, exudation, and edema [71]. Visual acuities were stable or improved in most patients (86%).

### 4.3.3. Justification

Working Group members agree that the reduction in disease recurrence when TTT is used as an adjunct to brachytherapy outweighs the treatment-related complications.

*4.4. Recommendations for Genetic Prognostic Testing*

- All patients should be offered a fine needle aspiration prior to treatment/at the time of treatment to obtain tumour genetic material for GEP or monosomy 3 and 8 testing to

provide information on survival prognosis. (Level of evidence: III [76,77] IV [78–81], Strength of recommendation: B).

### 4.4.1. Qualifying Statements

In Alberta, the commercially available GEP test is used instead of monosomy 3 testing because of the superior biopsy yields and prognostication [76].

### 4.4.2. Key Evidence

Several retrospective studies [79,80] and a nonrandomized case series [81] have shown that in uveal melanoma, abnormalities of chromosomes 3 and 8 are significant predictors of recurrence-free survival (RFS) and overall survival (OS). However, a prospective study evaluating the prognostic performance of a GEP assay showed that it effectively classified 97.2% of patients as being class 1 (low metastatic risk) or class 2 (high metastatic risk) [76]. While there was an association between GEP class 2 and monosomy 3, 20.8% were discordant for GEP and chromosome 3 status, among which GEP showed better prognostic accuracy (log-rank test, $p = 0.0001$). At 3 years follow-up, the net reclassification improvement of GEP or Tumour-Node-Metastasis classification was 0.43 ($p = 0.004$) over chromosome 3 status. Since the initial studies, tumour size has also been found to be an independent risk factor for metastasis [58,82–84].

### 4.4.3. Justification

Patients have reported that prognostic genetic testing in uveal melanoma helps to ease stress and support a more realistic risk perception [85]. Additionally, GEP or monosomy 3 and 8 testing is used in clinical practice to assist with eligibility for clinical trials and to guide appropriate follow-up for patients at high risk of developing distant metastases.

### *4.5. Recommendations for Surveillance Following Definitive Local Therapy*

- In patients who are currently disease-free but who would qualify for treatment should metastases develop, surveillance should be offered that may consist of history and physical examination, chemistry, and imaging based on patient risk factors. (Level of evidence: V, Strength of recommendation: B)
- For lower-risk patients, including those with GEP class 1 or disomy 3 (monosomy 3 negative or undetected) or patients with no genetic assessment and tumour ≤ 9 mm thick/12 mm in maximal basal dimension should have a physical examination and liver US once a year, for up to 10 years. Follow-up may be transitioned to a family physician at 5 years. (Level of evidence: V, Strength of recommendation: B)
- For higher-risk patients, including those with GEP class 2, monosomy 3 (monosomy 3 positive or detected), or tumours >9 mm thick/12 mm largest basal dimension without a genetic assessment should have a physical examination once a year, and imaging every 6 months alternating between liver US and liver MRI for 10 years. If limited by body habitus, consideration for other imaging modalities should be given. Follow-up care may be transitioned to a general practitioner at 5–10 years. As improvements in our ability to predict late metastasis evolve, and treatments of metastasis become more effective, follow-up recommendations are likely to change. (Level of evidence: V, Strength of recommendation: B)
- Long-term ocular evaluations to rule out local treatment failure and treatable radiation complications are recommended. (Level of evidence: V, Strength of recommendation: B)

### 4.5.1. Key Evidence

A small retrospective study evaluating trends in LFTs before detection of liver metastases from uveal melanoma reported that at the time of diagnosis of liver metastases by imaging, 50% of patients had at least one abnormal LFT compared to 5% of the control group [86]. In this study, alkaline-phosphatase (ALP) and LDH were the most predictive tests. LDH and aspartate-aminotransferase were already predictive at 80% of the upper

normal limit, while ALP and gamma-glutamyltransferase were most predictive at the upper normal limit. The authors concluded that monitoring changes in select LFTs (even within normal limits) can help predict metastatic uveal melanoma.

A retrospective study evaluating screening tests and their value in detecting metastatic disease at presentation reported that liver ultrasonography has 100% specificity but only 14% sensitivity [87]. Thus, the use of US in the follow-up of high-risk patients should complement other more sensitive tests.

Several small, prospective, and retrospective studies have looked at the use of imaging modalities in detecting metastases, mostly in the liver, at follow-up [30,31,88–91]. In the largest of these studies, a single-arm prospective cohort of 188 high-risk patients, 6-monthly hepatic MRI revealed metastases before symptoms developed in 92% of patients who developed systemic disease, which led to 14% of patients undergoing liver resection and surviving for at least a year after [88].

Other cancer organizations recommend similar systemic imaging, plus or minus blood tests based on risk stratification by genetic testing, plus or minus tumour size and histology at presentation [24,92].

Several prospective and retrospective studies report an increased risk of cutaneous melanoma following a diagnosis of uveal melanoma [88,93,94]. This risk varies significantly between studies though and may be related to increased surveillance. Thus, universal screening for cutaneous melanoma of these individuals is not currently recommended unless significant risk factors for cutaneous melanoma are present, in which case referral to a dermatologist for baseline total skin examination should be considered.

### 4.5.2. Justification

There is no high-level evidence to inform the best way to monitor patients who have undergone treatment for uveal melanoma. However, the Working Group recommends follow-up for high-risk patients because surgical resection, ablation, and immunotherapy may improve survival [33,95,96].

### *4.6. Recommendations for the Use of Systemic Therapy as Adjuvant to Local Therapies for High-Risk Patients*

- High-risk patients (based on tumour size and molecular testing) should be considered for clinical trials investigating the safety and efficacy of systemic therapy as adjuvant treatment to local therapies where possible. (Level of evidence: IV [97], Strength of recommendation: C)

### 4.6.1. Qualifying Statements

High-risk patients have any of the following: monosomy 3, chromosome 8 gain, GEP class 2, >12 mm in basal dimension, or >9 mm thick.

### 4.6.2. Key Evidence

No clinical trials support the use of adjuvant systemic therapy in high-risk patients. A retrospective cohort study compared OS in high-risk patients with uveal melanoma who received adjuvant sunitinib with institutional controls. At a median follow-up of 52.7 months, 51 deaths including 14 (26%) in the sunitinib group and 37 (50%) among the controls were reported. Based on univariate analysis, patients treated with sunitinib group had longer OS (hazard ratio [HR], 0.53; *p* = 0.041) [97].

### 4.6.3. Justification

At present, there are no Health Canada-approved agents for adjuvant (post-operative/ post-brachytherapy) use for patients. Given the absence of high-level evidence, enrollment in a clinical trial where possible is the most reasonable management strategy for patients at high risk of developing distant metastases.

*4.7. Recommendations for the Management of Patients with Metastatic Disease*

- All patients should have genotyping assay on whole blood for the presence of human leukocyte antigen (HLA)-A*02:01. Tebentafusp may be considered in the first-line setting for HLA-A*02:01-positive patients with unresectable or metastatic disease. (Level of evidence: I [98], Strength of recommendation: A)
- When possible, enrollment in a clinical trial is recommended. (Level of evidence: V, Strength of recommendation: B)
- Combined immunotherapy with ipilimumab/nivolumab or single-agent immunotherapy with nivolumab or pembrolizumab may be offered to patients with metastatic disease after discussion about lower effectiveness compared to patients with cutaneous melanoma. (Level of evidence: ipilimumab/nivolumab II [99], nivolumab II [100], pembrolizumab III [101], Strength of recommendation: B)
- Outside of a clinical trial, the routine use of palliative cytotoxic chemotherapy is not recommended. (Level of evidence: I [102] II [103–116], Strength of recommendation: D)
- Surgical resection of solitary/oligo liver metastasis may offer benefit in highly selected patients. (Level of evidence: IV [117–119], Strength of recommendation: C)
- Ablation modalities have also been used to treat patients with metastatic uveal melanoma including thermal ablation, radiofrequency ablation, and radioembolization. (Level of evidence: II [120], IV [95,121], Strength of recommendation: B)

### 4.7.1. Qualifying Statements

Health Canada approved tebentafusp for the treatment of unresectable or metastatic uveal melanoma in June 2022, and in December, a positive recommendation was made by the Canadian Agency for Drugs and Technologies in Health (CADTH).

In general, surgery is a preferred option in a fit patient with a solitary metastasis or oligo-metastatic disease amendable to resection. Most patients with metastatic disease have diffuse involvement of the liver and are not candidates for surgical resection.

There have been several liver-directed therapies that have been investigated in the metastatic uveal setting, both randomized and nonrandomized. The two randomized studies, FOCUS and SCANDIUM, utilized intrahepatic melphalan as the intervention and both demonstrated statistically significant improvements in PFS and overall response rate (ORR) as compared to best alternative care [122,123]. However, the OS data on these studies are still pending and these invasive procedures are typically only available at specialized tertiary cancer centres.

### 4.7.2. Key Evidence

Systemic and liver-directed chemotherapy for metastatic uveal melanoma has been largely modeled after chemotherapies for cutaneous melanoma. However, multiple phase II studies [103–116] and one phase III trial [102] have failed to demonstrate clinical efficacy in any of the tested single agents or combinations of agents, including, but not limited to temozolomide, dacarbazine/treosulfan, gemcitabine/treosulfan, sorafenib/carboplatin/paclitaxel, docosahexaenoic acid-paclitaxel, transarterial chemoembolization (TACE) using cisplatin, and hepatic intra-arterial treatment with fotemustine.

The use of receptor tyrosine kinase inhibitors (TKIs), including the MEK inhibitors selumetinib and trametinib and the c-KIT (CD117) inhibitor sunitinib, has also been studied in patients with metastatic uveal melanoma [124–128]. Resistance to these agents develops in a matter of months. While modest clinical activity with the use of these agents has been reported, none have yet been shown to improve OS [129]. Clinicians and patients who decide to use targeted therapies in the metastatic setting should understand that treatment-related toxicities may be significant and a detriment to quality of life.

A review was published in 2020 of the most important prospective and retrospective analyses investigating immune checkpoint blockade in patients with metastatic uveal melanoma [130]. Among studies using ipilimumab, the CTLA-4 inhibitor, the largest prospective study reported an ORR of 0% and a median OS of 6.8 months [131], while the

largest retrospective study reported ORR of 4.8% and a median OS of 6.0 months [132]. Among studies utilizing anti-PD-1 agents, the largest prospective study reported ORR of 5.8% and median OS of 11 months for patients on nivolumab [133], while the largest retrospective study reported ORR of 7.0% and a median OS of 10.3 months for patients on pembrolizumab [134]. Nivolumab and pembrolizumab both demonstrate activity in prospective, noncomparative phase II clinical trials [100,101].

Several studies have evaluated the effect of combined anti-CTLA-4/PD-1 immune checkpoint blockades. In a phase II clinical trial utilizing the combination of nivolimumab with ipilimumab, an ORR of 18% and median OS of nearly two years was reported [99]. One retrospective study reported ORR of 15.6% (3.1% complete response [CR]) and a median OS of 16.1 months [135] while another reported ORR of 11.0% (1% CR) and a median OS of 15.0 months [136]. The longest median OS in the literature is 18.9 months for combination ipilimumab and nivolumab, with an ORR of 21.0%, albeit with a small sample size [134]. A more recent study followed 30 metastatic uveal melanoma patients treated with ICI. The study had four patients survive for more than 5 years, all of whom received anti-CTLA-4 and anti-PD1, either sequentially or in combination.

Most recently, data from a randomized phase III clinical trial confirm a survival advantage for HLAA*02:01-positive adult patients treated with tebentafusp [98] When compared with investigator's choice of therapy (including ipilimumab, pembrolizumab, or dacarbazine chemotherapy), treatment with tebentafusp improved OS (HR 0.51) with a one-year OS rate of 73% and median OS of 22 months. This survival benefit with tebentafusp was confirmed with a longer follow-up of 36 months [137]. Median OS was 21.6 months (HR 0.68) in patients treated with tebentafusp and 16.9 months in patients treated with investigator's choice of therapy. Estimated 3-year survival was 27% in the tebentafusp group and 18% in the control group. The main limitation of tebentafusp is that approximately 50% of patients are ineligible for treatment based on their HLA status. In high-risk patients, it may be prudent to conduct early testing for HLA to efficiently assess whether tebentafusp or immunotherapy is the optimal initial treatment in the event of metastases development.

There are some data to suggest that resection of uveal melanoma liver metastases may prolong survival [119], including a single-arm prospective study among twelve patients who achieved a 5-year RFS of 15.6% and a 5-year OS of 53.3% following complete resection [138]. Retrospective data suggest that resection of liver metastases is associated with an approximately 3.7-fold increase in median survival, as compared to no surgery [117,118,139,140].

Surgical resection plus chemotherapy may benefit patients with metastatic disease. A prospective study of aggressive surgery (i.e., removal of as much liver disease as possible) and intra-arterial chemotherapy for 6 months (fotemustine and/or DTIC-platinum) among patients with liver metastases from uveal melanoma showed that complete macroscopic or R0 resection was possible in 27.5% and significant tumour reduction in 49.3% [141]. Median OS was 10 months in patients who received complete treatment surgery plus chemotherapy, and curative resection improved the median OS to 22 months (*p* < 0.001). Surgical or ablative therapy for uveal liver metastases is appropriate for a small minority of highly selected patients. Reports are anecdotal and differentiating therapeutic effect from selection and lead-time bias is not yet possible. A review of studies published since 2000 for each of the key treatment modalities available to patients with hepatic metastases from ocular melanoma concluded that only patients with limited metastases who can be rendered surgically free of disease should be considered for hepatic resections [142]. Incomplete resections appear to put patients at risk of surgical complications without any clear survival benefit. With the recent advances in systemic therapy, intra-arterial chemotherapy is no longer used after surgical resection.

Ablation modalities that have been used to treat patients with metastatic uveal melanoma include thermal ablation, radiofrequency ablation, and radioembolization. Although liver resection remains the gold standard, thermal ablation has the advantage of

sparing liver parenchyma, as well as providing a minimally invasive outpatient procedure [121,143]. In a phase II trial of radioembolization for the treatment of uveal melanoma hepatic metastasis, treatment-naïve patients achieved a median OS of 18.5 months with a 1-year survival of approximately 61% [120]. Participants treated with radioembolization in whom prior immunoembolization treatment failed achieved a median OS of 19.2 months with a 1-year survival of approximately 70%. A retrospective review of patients with liver metastasis from ocular melanoma who underwent surgery and/or radiofrequency ablation revealed that half of the patients had all metastatic liver lesions addressed [95]. The median survival of patients who underwent surgery alone or in combination with radiofrequency ablation to address all liver lesions was 46 months.

### 4.7.3. Justification

Tebentafusp is the first systemic treatment to show a survival benefit among patients with metastatic uveal melanoma. Emerging data suggest that combined immune checkpoint blockade may be superior to anti-PD1 or anti-CTLA-4 monotherapy, although there are limitations in the current data, including small sample sizes, potential selection bias, and a lack of clinical trials with comparative study design.

## 5. Discussion

This updated guideline outlines current best practices in the management of uveal melanoma with several key changes since our original publication in 2016. In addition to adding levels of evidence and strength of recommendations, changes include new and modified recommendations for provider training requirements, diagnostic imaging for the detection of metastases, neo-adjuvant pre-enucleation radiotherapy, intravitreal anti-vascular endothelial growth factor agents for radiation retinopathy, genetic prognostic testing, surveillance following definitive therapy, and systemic therapy for patients with metastatic uveal melanoma.

The strength of this guideline lies in its evidence-based recommendations compiled by a large team of multidisciplinary specialists involved in the care of patients with uveal melanoma. The limitation of this guideline, evident in the incorporation of ad hoc recommendations, is that the field is evolving, and certain sections of this guideline will soon require updates to ensure continued relevance and accuracy.

## 6. Conclusions

This evidence-based clinical practice guideline includes two strongly recommended practices (Grade A), fourteen generally recommended practices (Grade B), eight optional practices (Grade C), one generally not recommended practice (Grade D), and one never-recommended practice (Grade E). These recommendations represent the current standard of care that is feasible to implement by Alberta clinicians. The role of immune checkpoint blockades and liver-directed treatment for uveal melanoma will continue to be areas of interest that are anticipated to benefit patients with metastatic uveal melanoma.

**Author Contributions:** E.W., writing—original draft preparation; A.D.M., B.S., C.T.-O., E.W., J.M. (John McWhae), J.M. (Jose Monzon), J.W., G.M. (Greg McKinnon), G.M. (Geetha Menon), M.L., M.H., T.C., T.G.S. and X.K., writing—review and editing. All authors have read and agreed to the published version of the manuscript.

**Funding:** The Article Processing Fee was funded through the generous support of the Alberta Cancer Foundation for Alberta's Provincial Tumour Teams.

**Institutional Review Board Statement:** Not applicable.

**Informed Consent Statement:** Not applicable.

**Data Availability Statement:** Data are contained within the article. Evidence tables are available upon request.

**Conflicts of Interest:** E.W. participates on the Castle Biosciences, Inc. Advisory Board (no financial renumeration or interest) outside the submitted work. J.W. participated in Immunocore's phase II tebentafusp trial (ClinicalTrials.gov number, NCT03070392).

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
