# Peer review of "Management of Uveal Melanoma: Updated Cancer Care Alberta Clinical Practice Guideline"

_curroncol, doi:10.3390/curroncol31010002_

Round 1
Reviewer 1 Report
Comments and Suggestions for Authors
Brief Summary:
This is an updated guideline on the management of patients with uveal melanoma by Cancer Care Alberta. The authors performed a literature review of PubMed articles from January 1, 2014 through March 2, 2021, with an ad hoc update of the Tebentafusp approval. They present clinical practice guidelines based on the level of evidence and strength of recommendation.
General Comments:
This is an important update to the previously published 2016 guidelines by the Cancer Care Alberta group. They have included the newly approved evidence on tebentafusp. They should include another ad hoc recommendation update regarding the recent approval of percutaneous hepatic perfusion. They should also include more information on other liver-directed treatment modalities in the metastatic treatment section. Risk stratification should also include PRAME testing and gene mutations.
Specific Comments:
Section 2.4 Ad Hoc Recommendation Update
Please include the recent approval of percutaneous hepatic perfusion based on the FOCUS trial (Zager et al. JCO 2022, DOI: 10.1200/JCO.2022.40.16_suppl.9510).
Section 4.1 Recommendations for Diagnosis and Work-Up
I would recommend including fine-needle aspiration biopsy as part of the work-up. As the authors discuss later in the paper, GEP and chromosomal testing provide important prognostic information and possible clinical trial options.
Section 4.1.1 Qualifying Statements
The authors recommend a PET-CT as part of the staging work-up. MRI abdomen should be considered in all patients as part of the baseline imaging for all of the reasons the authors later describe in section 4.5.1 line 391-396.
In addition complete blood count and liver function tests as part of the initial blood work, LDH should also be included. A baseline LDH level is useful to have so it can be monitored as part of surveillance.
Section 4.1.2 Key Evidence
The NCCN guidelines that is referenced (reference #25) should be updated to the most current version (Version 1.2023).
Section 4.5 Recommendations for Surveillance Following Definitive Local Therapy
GEP Class 1 has two sub-classifications (1A and 1B) which should be considered when risk stratifying patients. PRAME testing should also be used to further risk stratify GEP Class 1 and 2 patients.
· Field, M. G. et al. PRAME as an independent biomarker for metastasis in uveal melanoma. Clin. Cancer Res. 22, 1234–1242 (2016).
· Field, M. G. et al. Epigenetic reprogramming and aberrant expression of PRAME are associated with increased metastatic risk in class 1 and class 2 uveal melanomas. Oncotarget 7, 59209–59219 (2016).
· Castle Biosciences. Uveal melanoma: DecisionDx-PRAME. Castle Biosciences https://castlebiosciences.com/products/decisiondx-prame-testing/ (2023).
Genetic alterations in the genes encoding BAP1, EIFA1X, and SF3B1 can also be useful in identifying risk and recommending surveillance frequency. Some studies have reported tumors with SF3B1 mutations develop late metastasis (Yavuzyigitoglu et al. Ophthalmology 2016, https://doi.org/10.1016/j.ophtha.2016.01.023).
Section 4.6.1. Qualifying Statements
Line 418 – There is a spelling error for “chromosome”. It should also specifically be chromosome 8q gain.
Section 4.7 Recommendations for the Management of Patients with Metastatic Disease
The authors should include information on other liver-directed therapies including the recently approved percutaneous hepatic perfusion (FOCUS trial, Zager et al. JCO 2022, DOI: 10.1200/JCO.2022.40.16_suppl.9510). Other modalities have also shown benefit and should also be considered and included:
· chemoembolization with BCNU (Gonsalves et al. AJR Am J Roentgenol 2015, https://doi.org/10.1016/j.ophtha.2016.01.023)
· immunoembolization (Valsecci et al. J. Vascl. Interv. Radiol. 2015, https://doi.org/10.1016/j.jvir.2014.11.037)
· Isolated hepatic perfusion (IHP) in the SCANDIUM trial (Bagge et al. JCO 2022, DOI: 10.1200/JCO.2022.40.17_suppl.LBA9509)
Radioembolization is not necessarily superior to these other techniques (not compared head to head in trials). I do not agree with the last recommendation that “radioembolization is preferred if ablative techniques are used” (line 452-453). For patients with one or two liver lesions, thermal ablation and radiofrequency ablation are appropriate and spare the potential risks of radioembolization including hepatic fibrosis, radiation pneumonitis, radiation-induced cholangitis, abscess formation, etc (Riaz et al. Front Oncol 2014, https://doi.org/10.3389%2Ffonc.2014.00198).
Comments on the Quality of English Language
No issues with English language, one minor spelling error.
Author Response
Dear Reviewer 3,
On behalf of the entire team of authors, I would like to extend our appreciation for your thorough review of our uveal melanoma manuscript. Your insightful comments and constructive feedback have helped to refine our resubmission.
We have carefully considered each of your suggestions and made the following changes:
- In Section 2.4 Ad Hoc Recommendation Update: added that we considered the new data re. percutaneous hepatic perfusion of melphalan. The Working Group decided that while it's an attractive option given the reported outcomes in the FOCUS trial, because of its limited availability and immature overall survival data, it's premature for us to make a recommendation about its use.
- In Section 4.1 Recommendations for Diagnosis and Work-Up: we have clarified in our recommendation that all patients should be offered a fine needle aspiration prior to treatment to obtain tumour genetic material for GEP or monosomy 3 and 8 testing to provide information on survival prognosis.
- In section 4.1.1 Qualifying Statements: We agree that MRI abdomen should be considered in all patients and have included alternative staging options including CT and MRI. We have also included LDH as part of the initial blood work.
- In section 4.1.2 Key Evidence: We didn't update the NCCN reference to Version 1.2023 because the version we used to inform this publication was from 2020.
- In section 4.5 Surveillance Following Definitive Local Therapy: We agree that the literature has differentiated between class 1A and 1B, and that PRAME status has shown to be related to prognosis in several studies. But, both class 1B and 1A and PRAME status have not been evaluated in a prospective multi-centre study to date. The Collaborative Ocular Oncology Group 2 study is currently evaluating these in this manner, but they have not presented their findings yet. We feel uncomfortable mentioning these without this validation since it's highly likely that how we use 1A, 1B, and PRAME will differ once the COOG 2 validation study is completed.
- In section 4.6.1 Qualifying Statements: We fixed the spelling error.
- In section 4.7 Recommendations for the Management of Patients with Metastatic Disease: From the studies suggested that we consider including, only the FOCUS and SCANDIUM are randomized trials. We have highlighted those two trials and their statistically significant improvements in PFS and ORR as compared to best alternative care, but don't yet recommend its use given the pending OS data and limited availability at this time. We also decided to remove our last recommendation that radioembolization is preferred if ablative techniques are used in the metastatic setting. Finally, given the opportunity to make changes, to our first recommendation, we added that all patients should have genotyping on whole blood for the presence of human leukocyte antigen (HLA)-A*02:01 and in the Key Evidence highlighted the new 3-year OS data with tebentafusp.
Thank you once again for your expertise and valuable insights. We look forward to the possibility of seeing our improved manuscript published in Current Oncology.
Best regards,
Brae Surgeoner, on behalf of the team of authors.
Reviewer 2 Report
Comments and Suggestions for Authors
The manuscript by Weis et al. provides an update on the clinical practice for managing Uveal melanoma. The study is well-described and helpful.
Author Response
Dear Reviewer 2,
On behalf of the entire team of authors, I would like to express our sincere gratitude for taking the time to review our uveal melanoma manuscript. We look forward to the possibility of seeing our manuscript published in Current Oncology. Thank you.
Best regards,
Brae Surgeoner
Reviewer 3 Report
Comments and Suggestions for Authors
Dear authors,
I really appreciate your work, based on the results of research published in recent years. Your paper brings concise recommendations for good clinical practice in the menagement of uveal melanomas. Recommendations made on the basis of the opinions of experts from different specialties and combined in one article can be a help in approaching a patient with this malignant disease.
Author Response
Dear Reviewer 3,
On behalf of the entire team of authors, I would like to express our sincere gratitude for taking the time to review our uveal melanoma manuscript. We look forward to the possibility of seeing our manuscript published in Current Oncology. Thank you.
Best regards,
Brae Surgeoner